

**Vertical Ionosphere Delay Estimation using Zero Difference GPS**
**Phase Observation**
Ahmed Elsayed[1], Ahmed Sedeek[2], Mohamed Doma[3], Mostafa Rabah [4]
1. Faculty of Engineering, Menofia University, Egypt,
2. lecture at Higher Institute of Engineering and Technology, El-Behira, Egypt,
3. Associate Professor at Faculty of Engineering, Menofia University, Egypt,
4. Professor at Faculty of Engineering, Benha University, Egypt.
**Abstract**
An apparent delay is occurred in GPS signal due to both refraction and diffraction
caused by the atmosphere. The second region of the atmosphere is the ionosphere.
The ionosphere is significantly related to GPS and the refraction it causes in GPS
signal is considered one of the main source of errors which must be eliminated to
determine accurate positions. GPS receiver networks have been used for monitoring
the ionosphere for a long time.
the ionospheric delay is the most predominant of all the error sources. This delay is
a function of the total electron content (TEC). Because of the dispersive nature of
the ionosphere, one can estimate the ionospheric delay using the dual frequency
GPS.
In the current research our primary goal is applying Precise Point Positioning (PPP)
observation for accurate ionosphere error modeling, by estimating Ionosphere delay
using carrier phase observations from dual frequency GPS receiver. The proposed
algorithm was written using MATLAB.
The proposed Algorithm depends on the geometry-free carrier-phase observations
after detecting cycle slip to estimates the ionospheric delay using a spherical
ionospheric shell model, in which the vertical delays are described by means of a
zenith delay at the station position and latitudinal and longitudinal gradients.
geometry-free carrier-phase observations were applied to avoid unwanted effects
of pseudorange measurements, such as code multipath. The ionospheric
estimation in this algorithm is performed by means of sequential least-
squares adjustment.
Finally, an adaptable user interface MATLAB software are capable of estimating
ionosphere delay, ambiguity term and ionosphere gradient accurately.





## 1. Introduction

During the transmission of GPS signals from satellite to receiver, the signals propagate through the ionosphere so that the ionospheric delay is closely associated with GPS and is considered one of the main sources of errors in point positioning using GPS techniques, on the other hand GPS can be used as a sensor of the ionosphere and investigate its characteristics because of the global system coverage and the availability of multiple frequency data.

In this paper we used GPS receiver as a sensor of the ionosphere. The ionosphere is a dispersive medium, which means that the delay depends on the frequency of the signal. the first order effect of the ionosphere refraction could be eliminated mathematically by means of a linear combination of the signals on the two frequencies, because GPS signals are broadcast on more than one frequency. This combination is widely called the iono-free combination (Leandro, 2009).

Various methods were devised to calculate the ionospheric delay. These methods were based on spherical harmonic expansions in the global or regional scale (e.g. Schaer, 1999, and Wielgosz et al., 2003a). Local methods were based on two-dimensional Taylor series expansions (e.g. Komjathy, 1997, Jakobsen et al. 2010, Deng et al 2009, and Masaharu et al. 2013).

This paper is aimed to apply Precise Point Positioning (PPP) observation for accurate ionosphere error modeling, using carrier phase measurements the proposed algorithm was written using MATLAB.

## 2. Observations equations for carrier-phase measurements.

The observations of dual-frequency GPS receiver at any station consists of two codes and two carrier phase observations in RINEX format which were used for present model. The observations equations for carrier-phase measurements can be formulated as follows (Leandro, 2009; e.g. Sedeek et al., 2017):

$$\Phi = R + c(dT - dt) + T - I + \lambda N + pbr - pbs + hdr - hds + m + e \qquad (1)$$

Where $\Phi$, $R$, $C$, $dT$ and $dt$, $T$, $I$, $\gamma$, $N$, $\lambda$, $hdr$ and $hds$, $pbr$ and $pbs$ and $m$ are the carrier-phase measurements, in meter, the geometric distance between satellite and receiver antennas, in meters, the speed of light, in meters per second, the receiver and satellite clock errors, respectively, in seconds, the neutral troposphere delay, in meters, the ionosphere delay, in meters, the carrier-phase integer ambiguity, the carrier-phase wave length, in meters, the receiver and satellite carrier-phase hardware delays, respectively, in metric units, the receiver and satellite carrier-phase initial phase bias, respectively, in metric units, the carrier-phase multipath, in meters, respectively and $e$ is the un-modeled errors of carrier-phase measurements, in meters.



## 3. Ionospheric Delay Estimation by Geometry-Free Linear Combination of GPS Observables.

The geometry-free linear combination of GPS observations is classically used for ionospheric investigations. It can be obtained by subtracting simultaneous pseudo range (P1-P2 or C1-P2) or carrier phase observations (Φ1-Φ2). With this combination, the satellite − receiver geometrical range and all frequency independent biases are removed (Ciraolo et al., 2007). The ionospheric estimation is performed using the following model (Leandro,2009):

$$\phi_{GF} = \phi_{L1} - \phi_{L2} = (1 - \gamma)\mathrm{MF}\left(I_{v,0} + \nabla_\phi(\phi_P - \phi_0) + \nabla_\lambda(\lambda_P - \lambda_0)\right) + Nb'_{gf} \qquad (2)$$

where $\phi_{GF}$ is the geometry-free carrier-phase observation in length units, MF is the ionosphere mapping function, $I_{v,0}$ is the vertical ionospheric delay at the station position, $\nabla\phi$ and $\nabla\lambda$ are latitudinal and longitudinal gradients, respectively, $\phi_P$ and $\lambda_P$ are the geodetic latitude and longitude of the ionospheric piercing point, $\phi_0$ and $\lambda_0$ are the geodetic latitude and longitude of the station, $\gamma$ *is the* factor to convert the ionospheric delay from L1 to L2 frequency, unitless and $Nb'_{gf}$ is an ambiguity parameter which includes the carrier-phase integer ambiguity plus a collection of biases. The mapping function is based on a spherical ionospheric shell model as shown in Figure 1, and is computed according to (Leandro,2009):

$$\mathrm{MF} = \sqrt{1 - \left(\left(\frac{r}{(r+sh)}\right)\cos(e)\right)^2} \qquad (3)$$

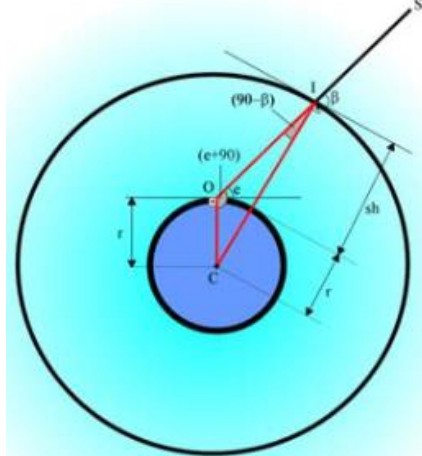

**Figure (1):** Elements of the ionospheric shell model (Leandro,2009).





where r is the mean radius of earth, $sh$ is the ionospheric shell height (default value
is 350 km), $\beta$ is the satellite elevation angle at the shell height piercing point, and $e$
is the elevation angle of satellite S from station O as seen in Figure (1).
To compute elevation and azimuth angle for any satellite (*e, Azim*), the receiver
position in Earth Centered Earth Fixed (ECEF) is converted to geodetic coordinate
$(\lambda, \varphi, z)$. Then, the satellite position coordinate $(x_s, y_s, z_s)$ from ECEF at the specified
epoch is interpolated from the IGS final orbits. The interpolated satellite position is
then transformed to a local coordinate frame, East, North, and Up (ENU) system.
The transferred ENU is used to calculate elevation and azimuth angles as follows
(Dahiraj, 2013 and Sedeek et al, 2017):
$$e = tan^{-1}\left(\frac{X_U}{\sqrt{X_N^2 + X_E^2}}\right) \qquad (4)$$
$$Azim = tan^{-1}\left(\frac{X_E}{X_N}\right) \qquad (5)$$
Where *e, Azim* are the elevation and azimuth angle of satellite at the receiver station
respectively and $X_E$, $X_N$, $X_U$ are the satellite position in local coordinate frame.
Usually, the ionosphere is assumed to be concentrated on a spherical shell located
at altitude (nominally taken as 350 km above Earth's surface. Ionospheric Pierce
Point is the intersection point between the satellite receiver line-of-sight, and the
ionosphere shell as shown in Figure (1).
IPP location can be computed by providing reference station coordinate $(\phi_0, \lambda_0)$,
then the geographic latitude and longitude of IPP can be computed according to
elevation and azimuth angle of satellite (Dahiraj, 2013). The offset angel between
the IPP and the receiver ($\psi$) is defined as the offset between the IPP and the user's
receiver. The elevation angle $\beta$ and the offset angel between the IPP and the receiver
$\Psi$ are computed as follow (El-Gizawy, 2003):
$$\beta = cos^{-1}\left(\left(\frac{r}{(r+sh)}\right)cos(e)\right) \qquad (6)$$
$$\Psi = \beta - e = cos^{-1}\left(\left(\frac{r}{(r+sh)}\right)cos(e)\right) - e \qquad (7)$$
Where $r$ and $sh$ are the mean radius of the spherical Earth and the height of IPP,
respectively. Given the user's receiver coordinates $(\phi_0, \lambda_0)$, and the offset angle $\Psi$,
the pierce point coordinates $(\phi_{IPP}, \lambda_{IPP})$ are then derived by the following
expressions (El-Gizawy, 2003):



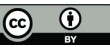

$\quad \phi_{IPP} = (\phi_r + \Psi \cos(\text{Azim}))$ $\hfill (8)$
$\quad \lambda_{IPP} = \left( \lambda_r + \frac{\Psi \sin(Azim)}{\cos(\phi_{IPP})} \right)$ $\hfill (9)$
The ionospheric estimation is performed by means of sequential least-squares
adjustment, where the parameters are the ionospheric model elements (vertical delay
and gradients) and the ambiguities as follows:
$\quad L=AX$ $\hfill (10)$
Where: $L$ is the vector of observations, $A$ is the design matrix, $X$ is unknown
parameters vector, and $P$ is weight matrix of observations.
$\quad X = (A_1^T.P_1.A_1 + A_2^T.P_2.A_2)^{-1}(A_1^T.P_1.L_1 + A_2^T.P_2.L_2)$ $\hfill (11)$
By using this system of equations, vertical ionospheric delay, latitudinal and
longitudinal gradients values at the station position are computed on an epoch by
epoch basis.
**4. Results and Discussions**
In the present contribution, to evaluate the performance of the proposed model,
numerical case-studies were performed on ten IGS stations. These stations are
shown in Figure (2).

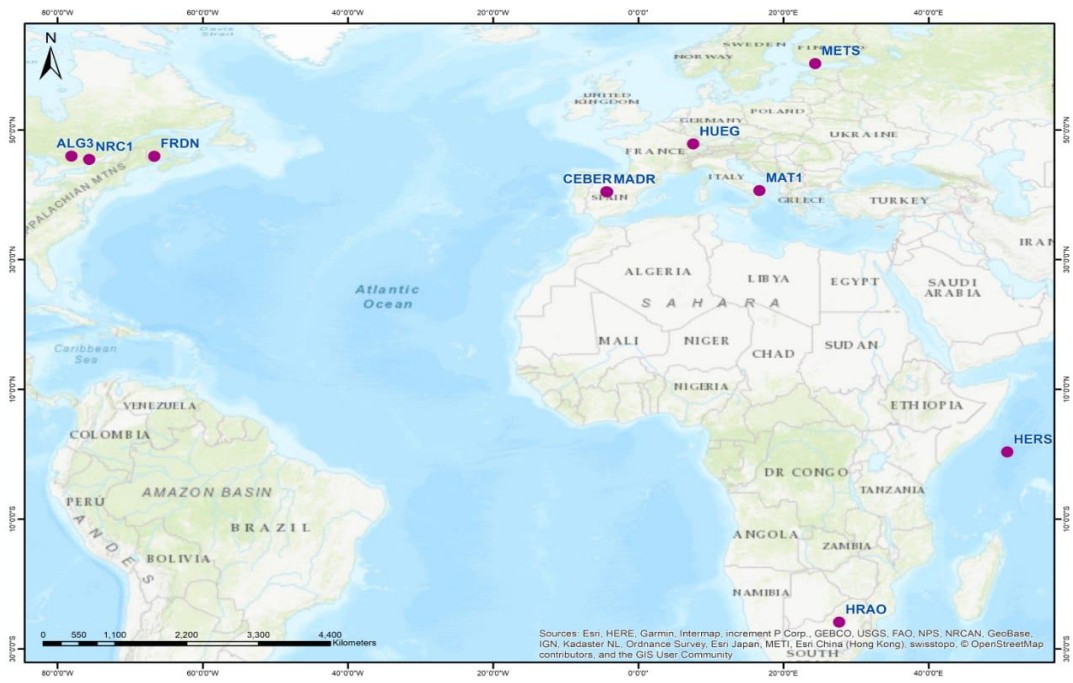

**Figure (2):** IGS stations which were used in this study.





The Ionosphere delay is estimated for observations of Doy 3, 2018 for these
stations and the results were compared with the results of the online version of the
GPS Analysis and Positioning Software (GAPS) as shown in the following
figures:

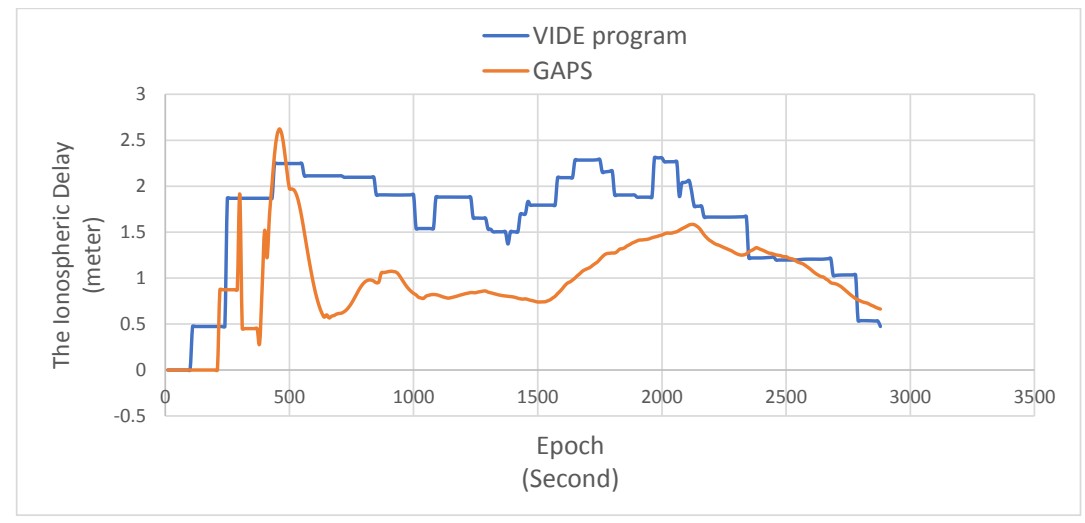


**Figure (3):** Vertical Ionosphere delay of ALGO station estimated by the VIDE program and GAPS of DOY 3, 2018.

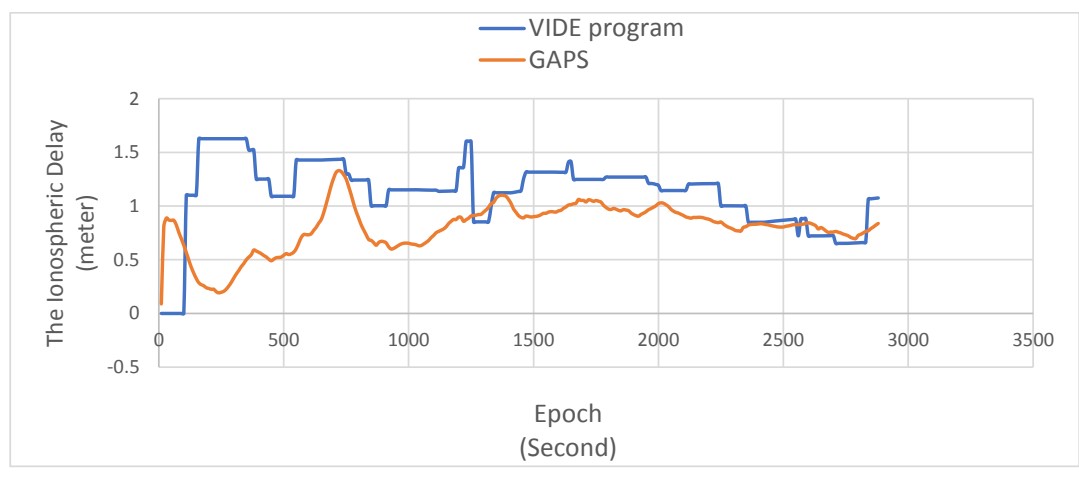


**Figure (4):** Vertical Ionosphere delay of CEBR station estimated by the VIDE program and GAPS of DOY 3, 2018.





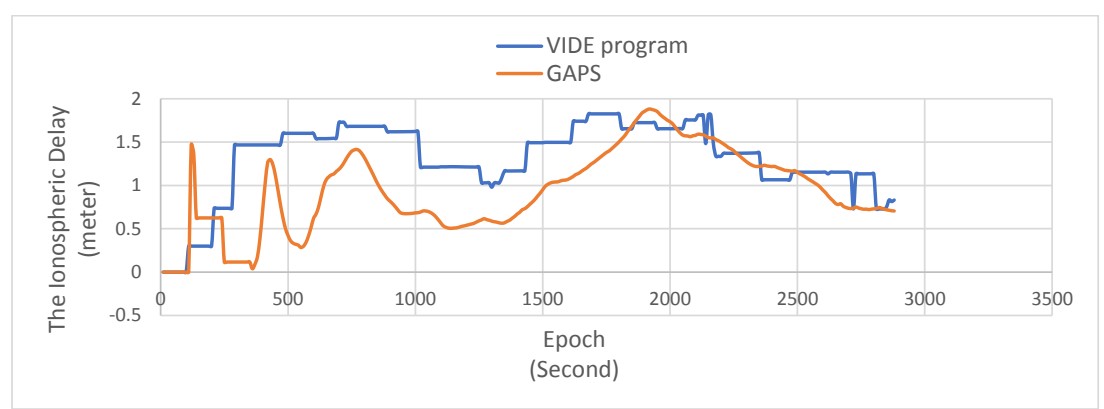


**Figure (5):** Vertical Ionosphere delay of FRDN station estimated by the VIDE program and GAPS of DOY 3, 2018.

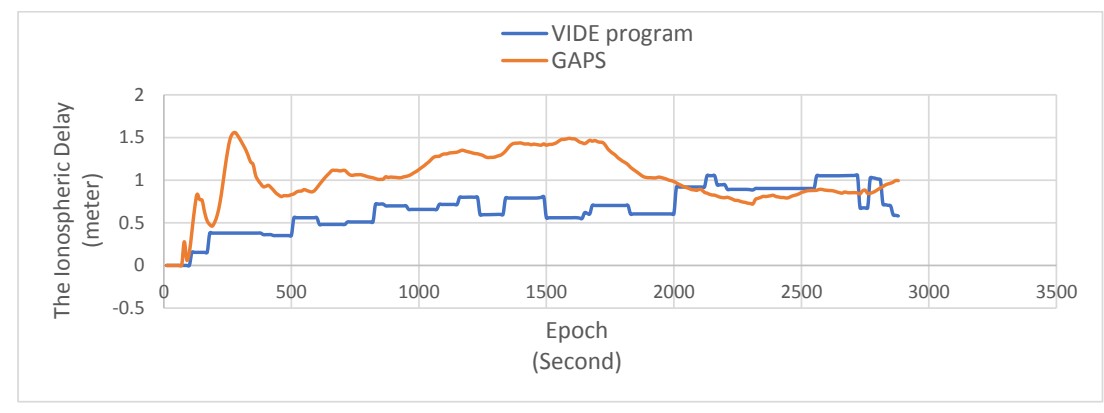


**Figure (6):** Vertical Ionosphere delay of HERS station estimated by the VIDE program and GAPS of DOY 3, 2018.

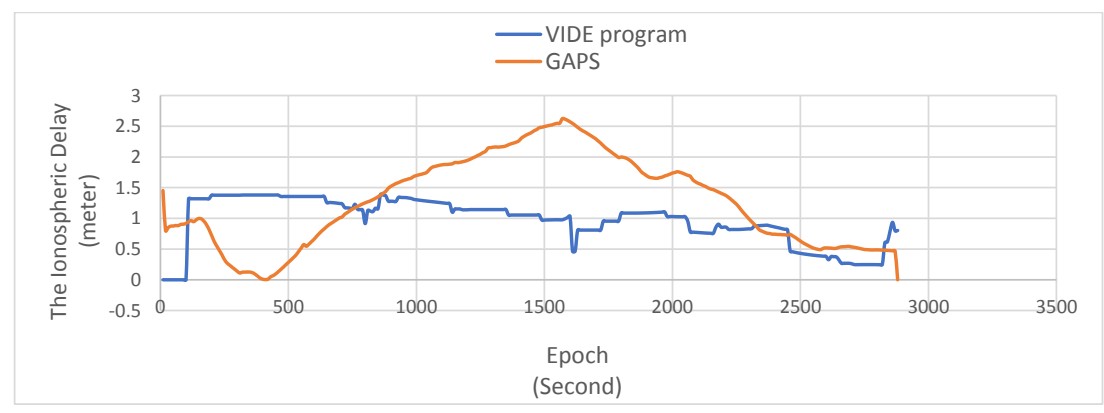


**Figure (7):** Vertical Ionosphere delay of HRAO station estimated by VIDE program and GAPS of DOY 3, 2018.





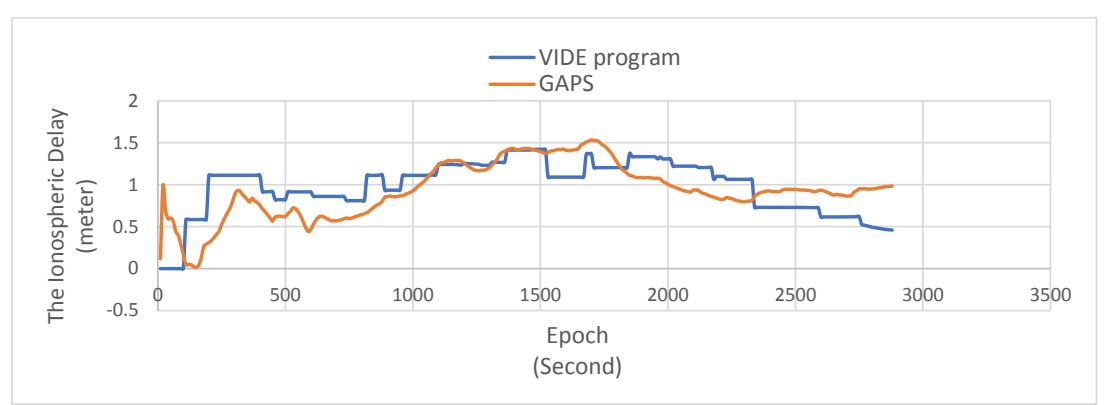


**Figure (8):** Vertical Ionosphere delay of HUEG station estimated by the VIDE program and GAPS of DOY 3, 2018.

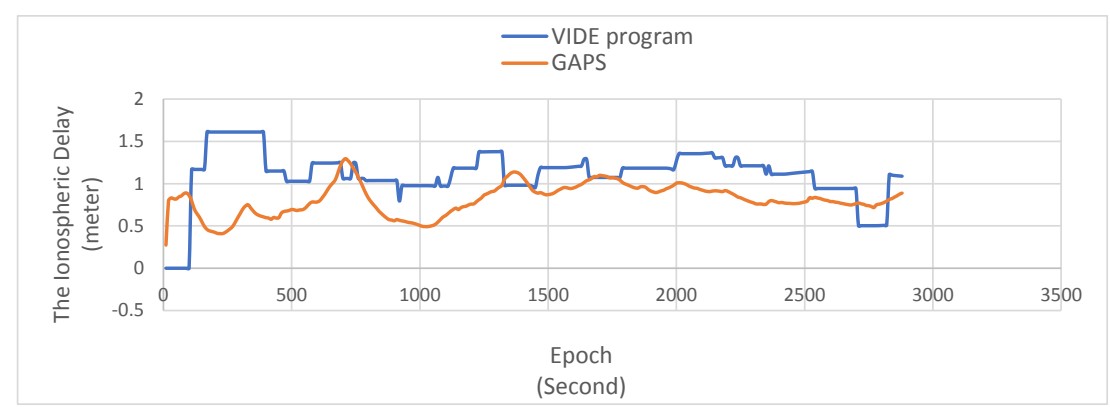


**Figure (9):** Vertical Ionosphere delay of MADR station estimated by VIDE program and GAPS of DOY 3, 2018.

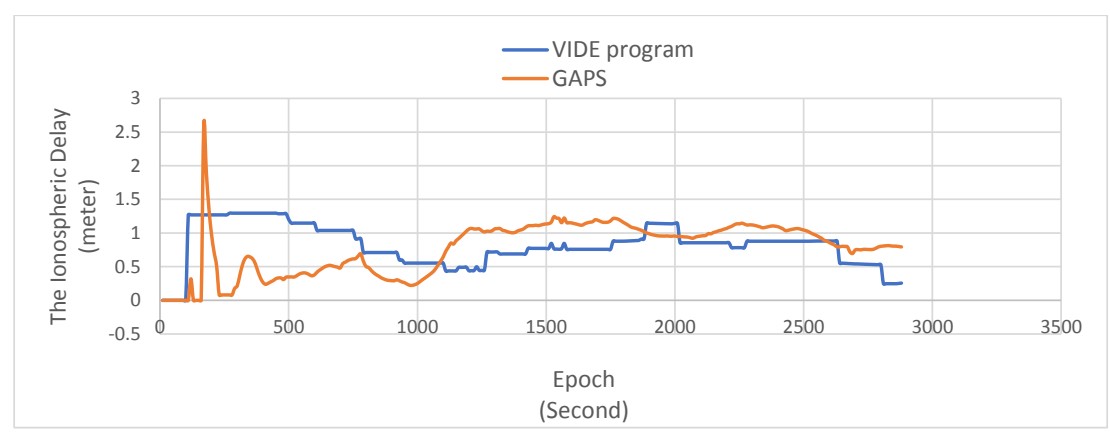


**Figure (10):** Vertical Ionosphere delay of MAT1 station estimated by VIDE program and GAPS of DOY 3, 2018.





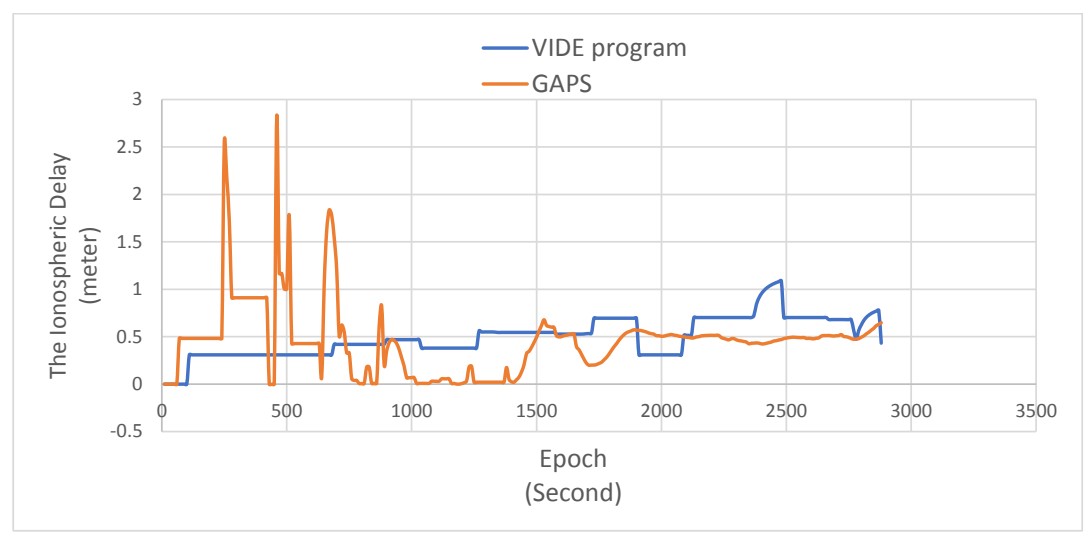


**Figure (11):** Vertical Ionosphere delay of METS station estimated by VIDE program and GAPS of DOY 3, 2018.

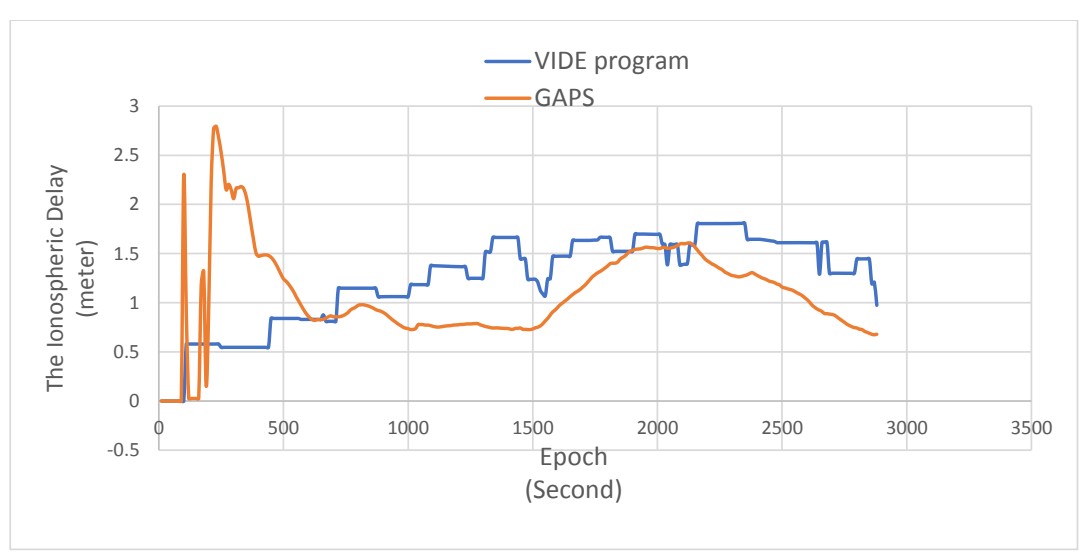


**Figure (12):** Vertical Ionosphere delay of NRC1 station estimated by VIDE program and GAPS of DOY 3, 2018.
Previous figures show a comparison of the ionospheric delays computed with the
proposed code and GAPS. This comparison shows how much the accuracy of this
study is good in terms of agreement of solutions provided by GAPS.



**Table (1):** The average Ionospheric Delay of each station of DOY 3, 2018 using the
Proposed code and GAPS.

| | Average Ionospheric Delay (m) | | | Average Ionospheric Delay (m) | |
|---|---|---|---|---|---|
| **Station** | **Proposed Code** | **GAPS** | **Station** | **Proposed Code** | **GAPS** |
| **ALGO** | 1.6205 | 0.9996 | **HUEG** | 0.9790 | 0.9328 |
| **CEBER** | 1.1203 | 0.7948 | **MADR** | 1.6838 | 0.8126 |
| **FRDN** | 1.3139 | 0.9387 | **MAT1** | 0.8189 | 0.9771 |
| **HERS** | 0.6594 | 1.0255 | **METS** | 0.4961 | 0.5106 |
| **HRAO** | 0.9681 | 1.2758 | **NRC1** | 1.2463 | 1.0848 |

**5.CONCLUSIONS**
We have overviewed an algorithm which can be used to estimate ionospheric delays
of GPS observations using single GPS receiver using a spherical ionospheric shell
model. This Algorithm depends on the geometry-free carrier-phase observations
after detecting cycle slip. The ionospheric estimation in this algorithm is performed
by means of Sequential least-squares adjustment. This study is performed on ten IGS
stations. Previous figures and table (1) show an agreement of the proposed code
results and values provided by GAPS. This procedure may be better than GAPS
because it can estimate the ionospheric delays each thirty seconds whereas GAPS
estimate the ionospheric delays each ten minutes.

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
