# Peer review of "Vertical Ionosphere Delay Estimation using Zero Difference GPS"

_Annales Geophysicae, 2018_

## Referee Comment (RC1) · Anonymous Referee #1 · 1 Dec 2018

The manuscripts present a model to estimate the Vertical Ionosphere Delays using Zero Difference GPS Phase Observation. The GPS Phase Observationobservations from ten IGS stationsare used. It is reported that the developed proposedprocedure may be better than GAPS because it can estimate the ionospheric delays each thirty seconds whereas GAPS estimate the ionospheric delays each ten minutes.However, I do not recommend the publication of their manuscript for the reasons given below. Nevertheless, I encourage the authors to submit a more mature manuscript at a later stage. Major 1. The description of proposed model needs more details. Do the observations from all stations or one station is considered to develop the model?,Where is the procedure for cycle slip detection by the model and what are the elements that considered to construct the design matrix A. 2. How the equation (10) elements are con-

sidered from the previous equationsand you are considering sequential least-squares (least-squares) for solving the unknowns and in equation (11)weight matrix isconsidered. 3. Regarding Figures, there is no explanation is given except figure description in once sentence. 4. VIDE program means what, VIDE program suddenly come in to figures, from my observation VIDE program means proposed Code. 5. The average Ionospheric Delay of each station of DOY 3, 2018 using the Proposed code is higher when compared to GAPS expect the locations of HERS, HRAO, MAT1 and METS. How the proposed model better than GAPS. Though you consider estimate the ionospheric delays each thirty seconds. Minor 1. Mention the web addresses or corresponding sources of the GAPS model. 2. When comparing observations and mode for several storms, it is needed to mention the station name at least in figures for quick understanding. 3. There are several minor typo and English mistakes. Example aberrate the terms, GPS, DOY,VIDE. Some time you are using Doy instead of DOY. 4. In Figure 2 : IGS station latitude and longitude not clearly visible. First mention the station station after use the station IDs. 5. In page 1 Line 15 "the ionospheric delay" the "T" must be the capital letter.Please see this king of simple mistakes.

---

## Author Comment (AC1) · 15 Dec 2018

**Replies to reviewer's comments**

**Comment to editor and reviewer**

The Authors are grateful to the editor and would like to thank the reviewer very much for his important comments that helped us to improve the original manuscript. We have responded to all comments and have revised the paper in light of them. Details of our responses to each comment are shown below.

**1- Major:**

| No. | Reviewer' comments | Authors' Responses |
|---|---|---|
| 1 | It is reported that the developed proposed procedure may be better than GAPS because it can estimate the ionospheric delays each thirty seconds whereas GAPS estimate the ionospheric delays each ten minutes. However, I do not recommend the publication of their manuscript for the reasons given below. | The publication of this manuscript aims to the following:

1- Studying the effect of ionosphere layer on GPS observations by introducing a MATLAB based software for GPS users capable on estimating the vertical ionospheric delay for single GPS receivers.

2- Investigating the performance of this software to be developed on real ground-based GPS observations by comparing its results with a well-known software "GAPS".

This new software provides results of ionospheric delay every thirty seconds. This time interval is a very reliable representation of the ionospheric effect on GPS observations. On the other hand, GAPS software provides these results every ten minutes. Thus, VIDE program can be considered more precise than GAPS software. |

| | | |
|---|---|---|
| 2 | The description of proposed model needs more details. Do the observations from all stations or one station is considered to develop the model? Where is the procedure for cycle slip detection by the model and what are the elements that considered to construct the design matrix A. | We estimate the ionospheric delays separately of each station. Moreover, we use ten case studies in different regions all over world in order to evaluate the effectiveness of our proposed software.

Melbourne-Wübbena Linear Combination of dual-band phase and code observables was used for detecting and repairing cycle slips in this current study as follows [Melbourne, 1985]:

$$L_6 = \lambda_{WL} N_{WL}$$
$$N_{WL} = (\Phi_1 - \Phi_2) - \frac{1}{\lambda_{WL}(f_1 + f_2)}(f_1 P_1 - f_2 P_2)$$

The elements that construct the design matrix A are as shown in Eq (2):
$(1 - \gamma)\mathrm{MF}$, $(1 - \gamma)\mathrm{MF}\,(\phi_P - \phi_0)$ and $(1 - \gamma)\mathrm{MF}\,(\lambda_P - \lambda_0)$ |
| 3 | How the equation (10) elements are considered from the previous equations and you are considering sequential least-squares (least-squares) for solving the unknowns and in equation (11) weight matrix is considered. | In Equation (11): L1 and L2 are the vectors of the first and the second groups of observations, respectively, $A_1$ and $A_2$ are the design matrices of the first and the second groups of observations, respectively, $P_1$ and $P_2$ are the weight matrices of the first and the second groups of observations, respectively, and $X$ is the unknown parameters vector.

**More details**:

are discussed in pages 189-190 in the following reference:

**Guochang Xu. (2007),** "GPS Theory, Algorithms and Applications", Library of Congress Control Number: 2007929855. ISBN second edition 978-3-540-72714-9 Springer Berlin Heidelberg New York.

7.2.1
*Least Squares Adjustment with Sequential Observation Groups*

Suppose one has two sequential observation equation systems

$V_1 = L_1 - A_1 X$ and $\qquad$ (7.11)
$V_2 = L_2 - A_2 X,$ $\qquad$ (7.12)

with weight matrices $P_1$ and $P_2$. These two equation systems are uncorrelated or independent and have the common unknown vector $X$. The combined problem can be represented as

$\binom{V_1}{V_2} = \binom{L_1}{L_2} - \binom{A_1}{A_2} X$ and $P = \begin{pmatrix} P_1 & 0 \\ 0 & P_2 \end{pmatrix}.$ $\qquad$ (7.13)

The least squares normal equation can be formed then as

$(A_1^T \ A_2^T)\begin{pmatrix} P_1 & 0 \\ 0 & P_2 \end{pmatrix}\binom{A_1}{A_2} X = (A_1^T \ A_2^T)\begin{pmatrix} P_1 & 0 \\ 0 & P_2 \end{pmatrix}\binom{L_1}{L_2}$

The solution is then

$X = (A_1^T P_1 A_1 + A_2^T P_2 A_2)^{-1}(A_1^T P_1 L_1 + A_2^T P_2 L_2).$ |

| | Reviewer' comments | Authors' Responses |
|---|---|---|
| 4 | Regarding Figures, there is no explanation is given except figure description in once sentence | A brief explanation of the results is mentioned in the line 162. Moreover, table (1) summarizes all the figures. |
| 5 | VIDE program means what, VIDE program suddenly come in to figures, from my observation VIDE program means proposed Code | Considered: (VIDE) is "Vertical Ionosphere Delay Estimation" which refers to the name of the new proposed software that was developed using MATLAB®. |
| 6 | The average Ionospheric Delay of each station of DOY 3, 2018 using the Proposed code is higher when compared to GAPS expect the locations of HERS, HRAO, MAT1 and METS. How the proposed model better than GAPS. Though you consider estimate the ionospheric delays each thirty seconds. | Because, it is the average through 24 hours. Moreover, this difference is high in the first epochs, then it is reduced gradually due to iterations using sequential least square method.

In addition, at the end of the day, the results of the ionospheric delay are almost the same.

Moreover, the standard deviation of results will be added in the revised paper. |

**2- Minor:**

| No. | Reviewer' comments | Authors' Responses |
|---|---|---|
| 1 | Mention the web addresses or corresponding sources of the GAPS model. | http://gaps.gge.unb.ca/submitadvanced.php |
| 2 | When comparing observations and mode for several storms, it is needed to mention the station name at least in figures for quick understanding. | The station name is already mentioned below each figure. |
| 3 | There are several minor typo and English mistakes. Example aberrate the terms, GPS, DOY, VIDE. Some time you are using Doy instead of DOY. | Considered, it will be corrected in the revised paper.

DOY refers to "Day of Year"

VIDE is "Vertical Ionosphere Delay Estimation" which refers to the name of the new proposed software.

GPS refers to "Global Positioning System". |

| | | Station | Longitude | Latitude |
|---|---|---|---|---|
| 4 | In Figure 2: IGS station latitude and longitude not clearly visible. First mention the station station after use the station IDs | ALG3 | -78.0727 ° | 45.9548 ° |
| | | CEBER | -4.3679 ° | 40.4534 ° |
| | | FRDN | -66.6599 ° | 45.9335 ° |
| | | HERS | 50.8673° | 0.3363° |
| | | HRAO | 27.6870 ° | - 25.8901 ° |
| | | HUEG | 7.5962 ° | 47.8340 ° |
| | | MADR | -4.2497 ° | 40.4292 ° |
| | | MAT1 | 16.7045 ° | 40.6491 ° |
| | | METS | 24.3953 ° | 60.2175 ° |
| | | NRC1 | - 75.6238 ° | 45.4542 ° |
| 5 | In page 1 Line 15 "the ionospheric delay" the "T" must be the capital letter. | Considered, it will be corrected in the revised paper. | | |

---

## Author Comment (AC2) · 21 Feb 2019

I am grateful to this journal and would like to thank editor very much for letting my manuscript under discussion in this valuable journal.

But i need to withdraw my manuscript due to some personal reasons, please.
* * *